**Cite this article:** de Oliveira Terceiro FE, Willems EP, Araújo A, Burkart JM. 2021 Monkey see, monkey feel? Marmoset reactions towards conspecifics' arousal. *R. Soc. Open Sci.* **8**: 211255. https://doi.org/10.1098/rsos.211255

behaviour/evolution

contagion of emotional arousal, consolation, empathy, sympathy, sympathetic concern

**Author for correspondence:**
Francisco Edvaldo de Oliveira Terceiro
e-mail: francisco.deoliveiraterceiro@uzh.ch

†Present address: Av. Sen. Salgado Filho, 3000-Candelária, Natal, Rio Grande do Norte 59064-741, Brazil.

# Monkey see, monkey feel? Marmoset reactions towards conspecifics' arousal

Francisco Edvaldo de Oliveira Terceiro[1,2,†], Erik P. Willems[2], Arrilton Araújo[1] and Judith M. Burkart[2]

[1]Department of Physiology and Behaviour, Universidade Federal do Rio Grande do Norte, PO Box 1511, Campus Universitário, 59078-970 Natal, Rio Grande do Norte, Brazil
[2]Department of Anthropology, Universität Zürich, Winterthurerstrasse 190, 8057 Zürich, Switzerland

FEdOT, 0000-0002-9071-604X; JMB, 0000-0002-6229-525X

Consolation has been observed in several species, including marmoset monkeys, but it is often unclear to what extent they are empathy-based. Marmosets perform well in at least two of three components of empathy-based consolation, namely understanding others and prosociality, but it is unknown to what extent they show matching with others. We, therefore, tested whether non-aroused individuals would become aroused themselves when encountering an aroused group member (indicated by piloerection of the tail). We found a robust contagion effect: group members were more likely to show piloerection themselves after having encountered an aroused versus relaxed conspecific. Moreover, group members offered consolation behaviours (affiliative approaches) towards the aroused fellow group members rather than the latter requesting it. Importantly, this pattern was shown by both aroused and non-aroused individuals, which suggests that they did not do this to reduce their own arousal but rather to console the individual in distress. We conclude that marmosets have all three components of empathy-based consolation. These results are in line with observations in another cooperative breeder, the prairie vole.

## 1. Introduction

After aggressive conflicts, several social species have been reported to engage in post-conflict affiliation, such as grooming and contact sitting, between former opponents (i.e. reconciliation behaviour) or between a former victim and non-involved bystanders (i.e. consolation behaviour [1]). Bystander affiliation toward former victims has for instance been shown in chimpanzees [1], bonobos [2] and Tonkean macaques [3], but not in Japanese macaques [4], baboons [5] or lemurs [6] (for a thorough review see [7]). Among

non-primates, it has also been proposed in canids [8], corvids [9,10] and elephants [11], pointing towards a broad phylogenetic distribution of consolation behaviour among big-brained social species. This is consistent with the idea that it is cognitively demanding.

As affiliative behaviours such as grooming universally decrease stress [12–14], post-conflict affiliation towards victims probably functions to console the victims by decreasing their stress levels. However, the bystander's goal must not necessarily be to reduce the stress of the victim and therefore need not involve concern for the latter's distress. In fact, its goal may be to reduce its own arousal because conflicts are often also stressful for uninvolved bystanders. Therefore, these group members may engage in affiliative behaviours not in order to console the former opponent, but to look for reassurance, as a coping mechanism to alleviate their own stress [15,16]. This is not implausible, as a major caveat of studying consolation in the context of post-conflict behaviour is that consolation is not necessarily an adaptive response. In particular when the threat of attack has not yet receded, lowering the arousal and thus alertness level in the victim can be detrimental.

Importantly, the motivation to alleviate the own distress is not mutually exclusive with the motivation to alleviate the victim's distress. As a consequence, disentangling these alternative motivations in a naturalistic context such as post-conflict behaviour is notoriously difficult. In particular, to exclude the reassurance seeking alternative, one would need to investigate bystanders who are not aroused themselves and nevertheless provide consolation behaviours, which is often not possible in naturalistic situations where conflicts often affect all group members to some degree. In a recent experimental approach with small-brained voles, this shortcoming was circumvented by inducing mild distress in a target animal in the absence of other group members [17], who, therefore, did not themselves experience the stressor. The authors found that prairie voles (*Microtus ochrogaster*), but not the closely related meadow voles (*Microtus pennsylvanicus*) performed consolatory behaviours towards the conspecifics in distress. Because prairie voles engage in cooperative breeding [18], unlike meadow voles [19], this suggests that social structure, rather than having a particularly large brain, may favour the evolutionary emergence of empathy-based consolation.

A crucial question is thus how affiliative behaviour directed at an individual in distress is motivated by a motivation to reduce the arousal in the other individual, or rather by the motivation to reduce one's own arousal. The latter is cognitively less demanding, whereas the former requires some form of understanding that the partner, rather than ego, is in distress, and is thus probably driven by an empathy-based mechanism such as sympathetic concern [17]. Sympathetic concern, i.e. an emotional response and concern about another's state, including attempts to ameliorate this state (de Waal [20], see also [15] for an in-depth discussion of the use of this term in different disciplines, and its relation to empathy), is increasingly seen as a multi-component concept, as highlighted in the combination model [15,21]. This model emphasizes three main components, namely *matching with others*, *understanding others* and *prosociality*. Each of these components may or may not be present in a given species, but all three of them are necessary to allow for empathy-based consolation. The *matching* component refers to the ability to mirror others' emotional state, such as in emotional contagion where the emotional state of a conspecific directly leads to the same emotional state in the observer. The *understanding others* component adds a cognitive dimension that helps an individual distinguish that this emotional state is not her own personal distress, but the partner's [22,23]. It thus supports the self–other distinction necessary to show affiliation in order to alleviate the distress of the partner, rather than ego. The *prosociality* component finally refers to a general concern about the well-being of others (i.e. proactive prosociality: [24]). Prosociality thus ensures that the negative state that is perceived in a partner through the matching mechanism, but correctly attributed to the partner and not to ego by the *understanding others* component, also motivates actions to alleviate the negative state of the partner.

This combination model of empathy and related concepts offers a novel route to investigating the evolutionary origin of empathy-based consolation, namely, to address these components separately in a given species. If a species provides evidence for all three of these components, and also engages in consolation-like behaviours, one can be increasingly confident that these behaviours are indeed mediated by sympathetic concern.

Our goal was to study how common marmosets (*Callithrix jacchus*) respond to others' arousal, to assess whether they engage in consolation behaviour, and how it may be motivated. Marmosets are promising candidates for several reasons. First, like prairie voles and humans, these small New World monkeys are cooperative breeders. They live in groups from 3 to 17 individuals composed of adult males and females, along with immatures. All group members help raising the immatures of the breeding female(s) [25]. Studies in wild populations have encountered groups with single breeding pairs as well as polygynous constellations [26–29]. Second, consolation-like behaviours have been

reported in this species [30]. One study showed that affiliation increased in an entire marmoset group when at least one member showed piloerection of the tail (which is a signal of arousal in this species [31]), but did not differentiate whether this was the result of a sympathetic concern or individual reassurance seeking. Third, marmosets are skilful in at least two of the three components highlighted in the combination model: (i) they are well known for their high levels of *proactive prosociality* [24,32–34]; (ii) they show a fundamental *understanding of others* as intentional agents (e.g. intention attribution crucially determines social learning, and they socially evaluate third-party interactions: [33–37]), visual perspective taking skills [32,38] and some differentiation between their own and others' needs [33,41–42]; (iii) marmosets show behavioural *matching* during social learning (true imitation: [43–45]). Moreover, daily playbacks of affiliative calls generated a temporary cultural style of high affiliation in marmosets [46], which is consistent with emotional contagion. Finally, they may be prone to contagion of emotional arousal since some arousal-related behaviours (gnawing, scent marking) cluster temporally in their social group [47,48]. However, this study could not exclude an important alternative to matching behaviours with others, namely that the animals were simultaneously but independently responding to some external stimulus.

Therefore, we studied consolation in marmoset monkeys, with a paradigm similar to the one used with the voles [17]. This would allow us to further assess (i) to what extent marmosets possess the third component of the combination model, namely the matching component, in the specific context of contagion of emotional arousal and (ii) whether they engage in consolation behaviour (i.e. establishing contact and direct affiliative behaviours specifically toward aroused individuals). The core of our paradigm is to produce an asymmetry in arousal between group members to address the alternative explanation of reassurance seeking. If consolation behaviour is directed from a non-aroused individual to an aroused partner, this alternative explanation can be excluded. We, therefore, either offered a high-quality food item to a single individual or showed the same food item without granting access to it. This food teasing elicits moderate arousal in marmosets as shown with thermography [31], which reliably coincides with piloerection of the tail. Once the remaining group members were allowed to return to the home enclosure, they thus encountered an aroused focal individual without being themselves aroused and without a source of arousal being present, creating the crucial asymmetric situation of arousal that would allow us to test for both the presence of contagion of emotional arousal and consolation.

We predicted that (i) if marmosets showed contagion of emotional arousal, the group members who encountered an aroused focal individual upon return would match the arousal of the focal individual and show piloerection of the tail themselves. Moreover, (ii) if marmosets engaged in consolation as suggested by the pattern of results in de Boer *et al*. [30], the establishment of friendly social contact should more likely be initiated by the group members when the focals were aroused than non-aroused. Likewise, the direction of affiliative behaviours should more likely go from group members towards the focal individual when the latter was aroused than when not. Importantly, to exclude the alternative explanation of reassurance seeking, these behaviour patterns have to be shown also by those group members who did not, as a result of contagion, show a piloerection of the tail themselves.

## 2. Methods

### 2.1. Subjects

We tested 16 animals from four captive groups. All subjects were previously habituated to the experimenter. Each group was composed of a breeding pair and two adult offspring (i.e. helpers) of this pair and thus could be composed of four classes of group members (i.e. sex-status classes: female breeders, female helpers, male breeders, male helpers). Two of our groups had a male and a female adult offspring, whereas the other two groups had two females. All groups were housed in indoor home enclosures ($1.8 \times 2.7 \times 2.4$ m) and had access to outdoor enclosures ($1.8 \times 3.7 \times 2.4$ m). Home enclosures were equipped with a sleeping box, a water dispenser, several wooden climbing structures, frequently changing enrichment devices, an infrared lamp and a mulch floor. The outdoor enclosures contained natural soil and vegetation, as well as wooden climbing structures. For more details on husbandry conditions see [31]. All animals were kept in accordance with Swiss legislation, husbandry licence 116 from the Kantonales Veterinäramt Zurich. All experiments were in concordance with the ethical regulations in Switzerland from the Kantonales Veterinäramt ZH (licence no. 223/16, degree of severity = 0).

## 2.2. Apparatus and procedure

We habituated all subjects to enter an experimental cage ($40 \times 40 \times 75$ cm) wrapped with black opaque cloth and placed inside their home enclosure, feeding them intermittently with mealworms through the front grid. Throughout 15 days, all individuals were put under this habituation protocol until all subjects would spend at least 5 min in the experimental cage focused on the experimenter and without signs of arousal. Thereupon, we again separated an individual randomly, the focal subject, in the experimental cage and implemented one of two experimental conditions: in the control condition, they received a piece of high-valued fruit (i.e. a whole grape) from the experimenter; in the test condition, the experimenter showed them a similar piece of food without handing it over to them, until they showed piloerection of tail as sign of arousal (teasing). The animal remained in the cage until it had consumed the grape (feeding condition) or until it showed piloerection of the tail (teasing condition) but no longer than 3 min. In the latter case, the experiment was repeated on a different day. During the implementation of the treatments, the remaining subjects (hereafter group members) were allowed to move to the outdoor enclosure and the access door was closed. Therefore, these group members (the potential consolators) would not witness what happened to the focal subject during the treatment. The returning group member's first data point was collected immediately after the sliding door of the experimental cage was open (time 0.00). The time elapsed between group members returning and focal subject release was a few seconds short. Thus, the arousal status for time 0.00 corresponds to their status upon returning. After the treatment, the group was reunited in the home enclosure by opening the door to the experimental cage, the experimenter left, and all behaviours were video recorded for 10 min (for a graphical representation of the procedure, see figure 1). The entire procedure, from the end of the manipulation until the start of the post-manipulation period, averaged no longer than 1 min. The piloerection of the focal (in the teasing condition) was checked if still present at the moment of release, during the experiment and again afterwards from the videos. If that was not the case the session was aborted.

The behaviour and arousal state of all individuals was later quantified from the video recordings. Behavioural recording started when the sliding door of the experimental cage was opened to release the subject back to its group and finished exactly 10 min later. The sliding door was only opened after all group members had returned from the outdoor enclosure. We tested each marmoset five times in each condition (total of 80 sessions for each condition). Per group, only one experimental session was run per day, and tests were performed four times per week.

## 2.3. Data coding

We used INTERACT for video analyses coding the behaviours defined as follows:

— Piloerection of the tail: tail fur in clear upwards position.
— Affiliative interaction: all interactions between the focal subject and the remaining group members that could be categorized as either co-feeding, food sharing, grooming or contact sitting as defined below:
— Co-feeding: two or more individuals eating the same piece of food at the same time.
— Food sharing: a change of food possession from one individual to the other.
— Grooming: a individual picks through and/or slowly brushes aside the fur of another individual using their hands.
— Contact sitting: two or more individuals sitting next to each other for at least 2 s.
— Establishing affiliative contact: Approach from one individual towards another group member immediately before engaging in one of the affiliative interactions defined above.

Throughout the entire 10 min experiment, we recorded every subject's piloerection status (piloerection present or absent) every 30 s, to track their arousal state over time. We also registered every affiliative interaction between the focal individual and all other group members, following the behavioural definitions detailed above during the same 10 min span as an event for co-feeding and food sharing and, as both duration and event, for the remaining affiliative behaviours. For establishing affiliative contact and affiliative interactions, we additionally registered which individual was being approached and which one was initiating the affiliative contact. in the case of mishaps during video recording (i.e. both cameras blocked by marmosets) obstructing the view at the moment of recording piloerection, that specific data point was removed from piloerection analysis. Ten per cent of the trials were

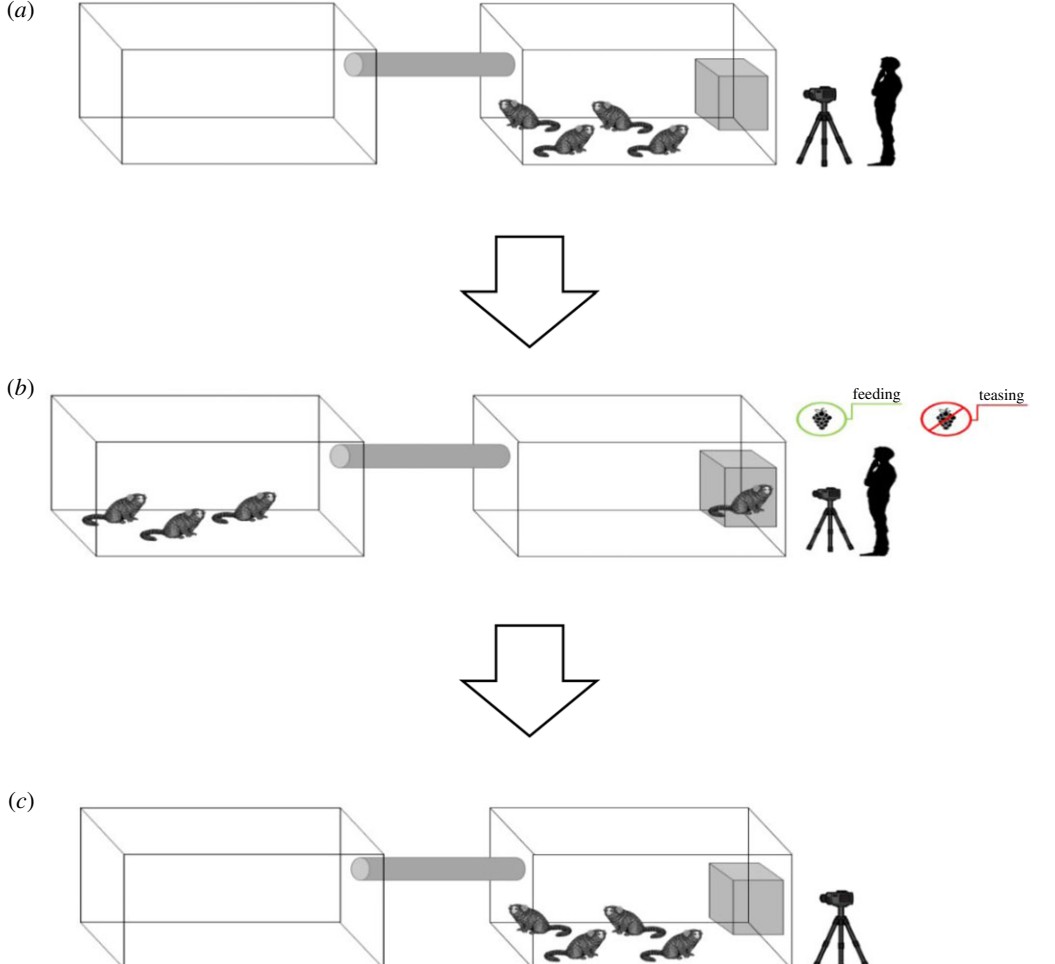

**Figure 1.** Experimental set-up in three steps. (*a*) Focal subject is separated in the experimental cage and the remaining group members go to the outside enclosure. (*b*) The experimenter displays food in one of two conditions (feeding/teasing). (*c*) Group reunited in home enclosure for behavioural recording (experimenter absent). Note that for better visibility, the size of the monkeys is disproportionately bigger relative to the enclosures than it is in reality.

randomly selected and coded by an independent coder. We used the intraclass correlation coefficient (ICC 3,1) to assess the reliability of the ratings. The independent coder reached a consistency of 0.92, 95% CI [0.78, 0.95] on time duration measures and inter-rater reliability of 0.95, 95% CI [0.91, 0.98] for behaviour measures (events).

## 2.4. Statistical analysis

We applied a binomial approach quantifying *piloerection* (*models 1* and *2*), as well as, *establishing the first affiliative contact* and *affiliative interaction* (*models 3* and *4*) and have later displayed these categories as numeric averages on our graphical representation in the results (for more detail see above Data coding section). In the Results section, we report the full versus null model comparison. We used 60 s bins instead of 30 s bins for piloerection to ascertain model convergence.

All statistical analyses were conducted in RStudio 1.1.463. Generalized linear mixed models (GLMM) were performed with the R function 'glmer' of the package 'lme4', and binomial as family. The models were built up based on our hypotheses, starting with a null model that only included the random effects (random intercept), adding all the fixed effects and their interactions gradually, and later perused with the Akaike information criterion (AIC). Model parameters were estimated using optimization algorithms in the package 'optimix', and *post hoc* pairwise comparisons were conducted using estimated marginal means from the 'emmeans' package (see also electronic supplementary material, methods A and table S1). Following our predictions (highlighted in italic), our statistical analyses were organized in three groups of tests.

### 2.4.1. Focal reactions to the experimental manipulation

Here, we analysed the focal subject's reaction to the experimental treatment. We used piloerection (yes/no) every 60 s as dependent variable, experimental condition (feeding/teasing), focal sex, focal status and time as independent variables (*model 1*). We used focal subject nested in group as our random factor.

### 2.4.2. Contagion of emotional arousal

Since the analyses of the focal subjects revealed that arousal effects from the treatment were only present in the first 3 min, we focused here on these first 3 min to evaluate arousal matching in the group members. We used the group member's piloerection (yes/no) as dependent variable (*model 2*) and focal tail condition, focal subject sex-status and group member sex-status as independent variables. Sex-status is a subject's category combining its sex (male/female) and breeding status (breeder/helper), thus providing four possible sex-status categories, i.e. female breeders, female helpers, male breeders, male helpers. We also used focal subject nested in group and group member nested in group as our random factors.

### 2.4.3. Consolatory behaviours

We analysed whether returning group members were more likely to establish affiliative contact with an aroused focal individual or vice versa (yes/no—dependent variable: *model 3*) and to have affiliative interactions with an aroused focal individual (yes/no—dependent variable: *model 4*). This distinction, although subtle, is necessary to disentangle possible differences between the individual seeking a partner and the individual undertaking an affiliative interaction (e.g. group member A moves towards focal B and focal B grooms group member A). For *model 3* (establishing the first affiliative contact) we used group member tail condition, focal subject tail condition and focal subject sex-status as independent variables, whereas for *model 4* (affiliative interactions) we used group member tail condition, focal subject tail condition and focal subject sex as independent variables. Group member and focal subject tail condition were defined as piloerection within the first 3 min of each session for at least one of our scans every 30 s, as a consequence of the results of our first two models. Since affiliative initiations and interactions rarely happened in the first 3 min, these analyses included the entire duration of 10 min. Moreover, to avoid capturing social interactions that may have unfolded during this period independently of the treatment, we only analysed the *first* social interaction from each session of the *first* group member to interact with the focal subject. Focal subject nested in group was used as random factor. However, this conservative procedure also led to a smaller and more unbalanced dataset in the analysis of consolatory behaviours (see also electronic supplementary material, methods A and figure S1). Therefore, we used AIC values to assess if sex-status, or one of its sub-parts (sex or status), was still the more informative grouping (see also electronic supplementary material, methods A and table S2) which resulted in our decision to use focal sex in *model 4*. For full statistical details on each model pairwise comparison see also tables S3–8 in our electronic supplementary material, results.

# 3. Results

## 3.1. Focal reactions to the experimental manipulation

We first quantified the response of the focal subjects to the experimental treatment, to ascertain that the test treatment indeed reliably led to higher arousal than the control treatment, and to investigate how long this effect would persist. We compared piloerection rates throughout the entire 10 min session comparing minute by minute between conditions and found that the focal subjects had higher piloerection rates in their first 3 min of the experimental condition ($\chi^2 = 1695.4$, d.f. = 21, $p < 0.0001$—*model 1* in table 1, see also electronic supplementary material, results B and table S3), whereas from minute 4 onwards piloerection was mostly absent in both conditions (figure 2).

Based on these results we focused for the contagion analysis on the first 3 min only. Moreover, since the split in aroused versus non-aroused focals did not perfectly coincide with the feeding versus teasing condition, for the further analyses we compared their response to aroused versus non-aroused focal subjects, regardless of condition. Since the latter contrast is what is visible to the returning group members, this is the more relevant one to investigate contagion of emotional arousal and consolatory behaviours.

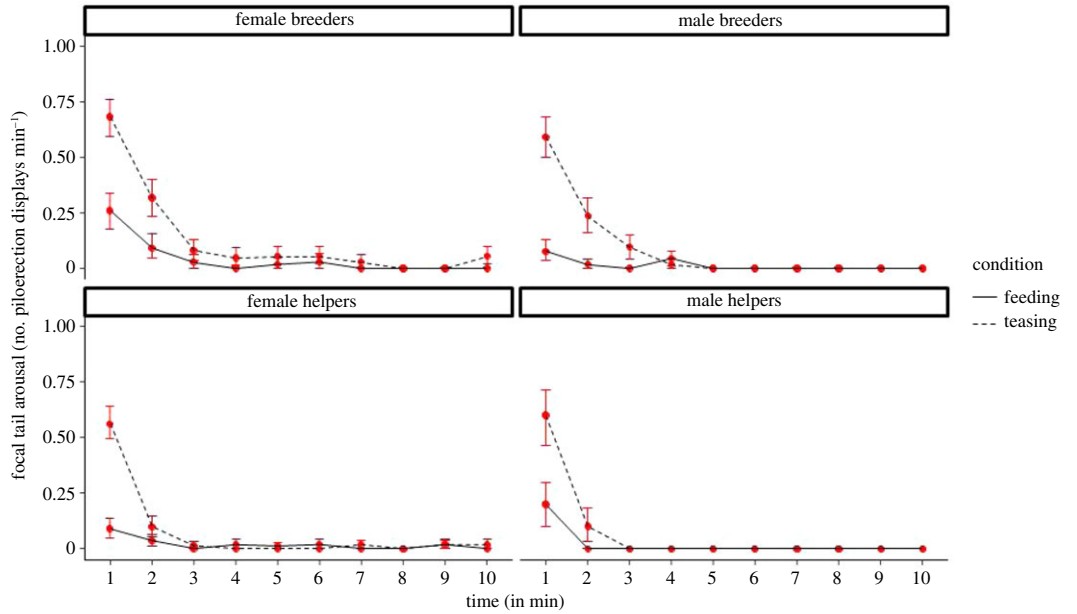

**Figure 2.** Focal piloerection of the focal subjects; number of times a subject displayed piloerection per minute, split up according to condition, time (minutes after reunion) and sex-status categories (estimated marginal means ± 1 s.e.m., $N = 16$). Full lines represent the feeding condition and dotted lines represent the teasing condition.

**Table 1.** ANOVA table on focal subject reaction to experimental procedure. Italics, $p < 0.005$.

| fixed factors | $\chi^2$ | d.f. | $p$-value |
|---|---|---|---|
| *model 1*: focal subject reaction to experimental procedure | | | |
| experimental condition | 194.460 | 1 | *<0.001* |
| focal sex | 1.423 | 1 | 0.233 |
| focal status | 1.780 | 1 | 0.182 |
| time in minutes | 701.903 | 9 | *<0.001* |
| experimental condition ∗ time in minutes | 67.081 | 9 | *<0.001* |
| full model ($R^2m = 0.895$; $R^2c = 0.930$) | 1695.4 | 21 | *<0.001* |

**Table 2.** ANOVA table on group members' reaction to aroused focal subjects in the first 3 min. Italics, $p < 0.005$.

| fixed factors | $\chi^2$ | d.f. | $p$-value |
|---|---|---|---|
| *model 2*: group member's tail condition | | | |
| focal tail condition | 157.040 | 1 | *<0.001* |
| focal subject sex-status | 9.904 | 3 | *0.019* |
| group member sex-status | 29.920 | 3 | *<0.001* |
| focal tail condition ∗ focal subject sex-status | 2.480 | 3 | 0.471 |
| full model ($R^2m = 0.324$; $R^2c = 0.549$) | 204.35 | 10 | *<0.001* |

## 3.2. Contagion of emotional arousal

To investigate whether the piloerection of the aroused focal individuals would be contagious and lead to piloerection in the other group members too (matching the arousal state), we analysed the group members' piloerection during the same first 3 min when they encountered the focal subject who was either aroused or relaxed. The group members display higher rates of piloerection after witnessing a focal subject who was aroused compared with a relaxed one ($\chi^2 = 204.35$, d.f. = 10, $p < 0.0001$—*model 2* in

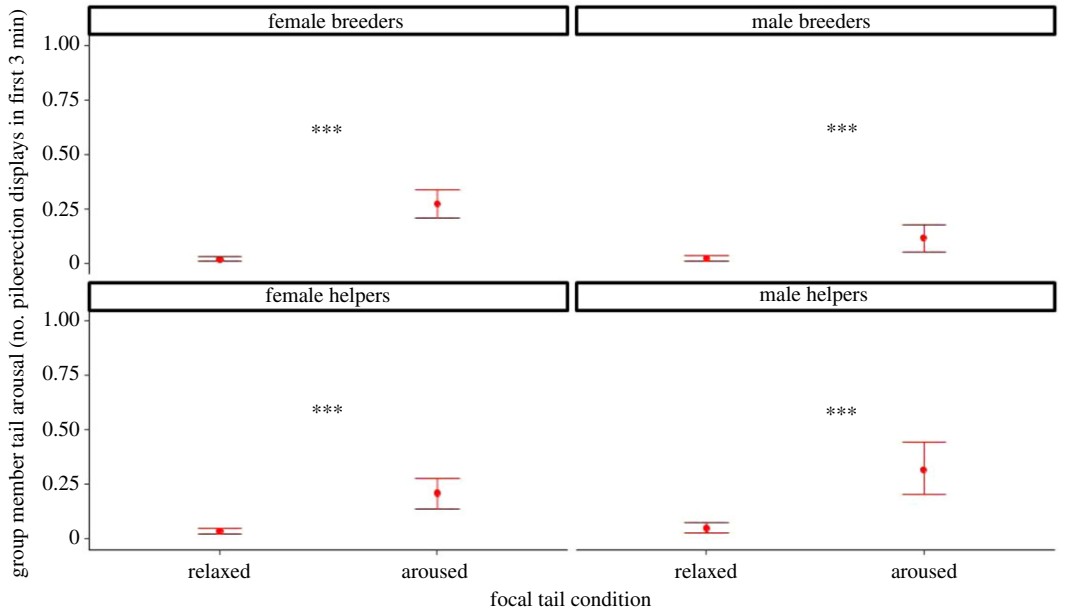

**Figure 3.** Group members' matching of piloerection; number of times a subject displayed piloerection in first 3 min on average, split up according to condition and focal subject sex-status categories. *Post hoc* pairwise tests ***: $p < 0.001$, (estimate marginal means $\pm$ 1 s.e.m., $N = 16$).

**Table 3.** ANOVA table on the probability that group members (rather than focals) are responsible for establishing the first affiliative contact. Italics, $p < 0.005$.

| fixed factors | $\chi^2$ | d.f. | *p*-value |
|---|---|---|---|
| *model 3*: establishing the first affiliative contact | | | |
| group member tail condition | 16.573 | 1 | *<0.001* |
| focal tail condition | 2.976 | 1 | 0.084 |
| focal subject sex-status | 1.023 | 1 | 0.796 |
| focal tail condition ∗ focal subject sex-status | 13.994 | 3 | *0.002* |
| full model ($R^2m = 0.368$; $R^2c = 0.801$) | 54.013 | 8 | *<0.001* |

table 2), and this response holds true regardless of the sex or status of the focal individual, although females elicited the strongest effect (figure 3; see also electronic supplementary material, results C and table S4 for pairwise comparisons), and also regardless of the group member's sex and status, even though female breeders were more responsive (see also electronic supplementary material, results C and table S5 for pairwise comparisons). Thus, returning group members reacted to aroused focal individuals by matching the arousal, and this matching was present regardless of whether the focal subject or the returning group member was a male or a female, and a helper or a breeder. A similar overall tendency was found in the same direction when we compared conditions (feeding versus teasing) instead of piloerection of the focal, especially among females (see also electronic supplementary material, results C and table S6 and figure S2).

## 3.3. Consolatory behaviours

First, we analysed whether the first affiliative contacts between focal individuals and group members were more likely initiated by the focal individuals or the group members. The full model predicted the data better than the null model (see table 3 for full model, $\chi^2 = 54.013$, d.f. = 8, $p < 0.001$). We found that for males as focal subjects, both breeders and helpers, these first affiliative contacts were more likely initiated by a group member than the focal when the focals were aroused (piloerection) rather than when they were relaxed (figure 4; see also electronic supplementary material, results D and table S7). Although we cannot perform a three-way interaction model with our current dataset

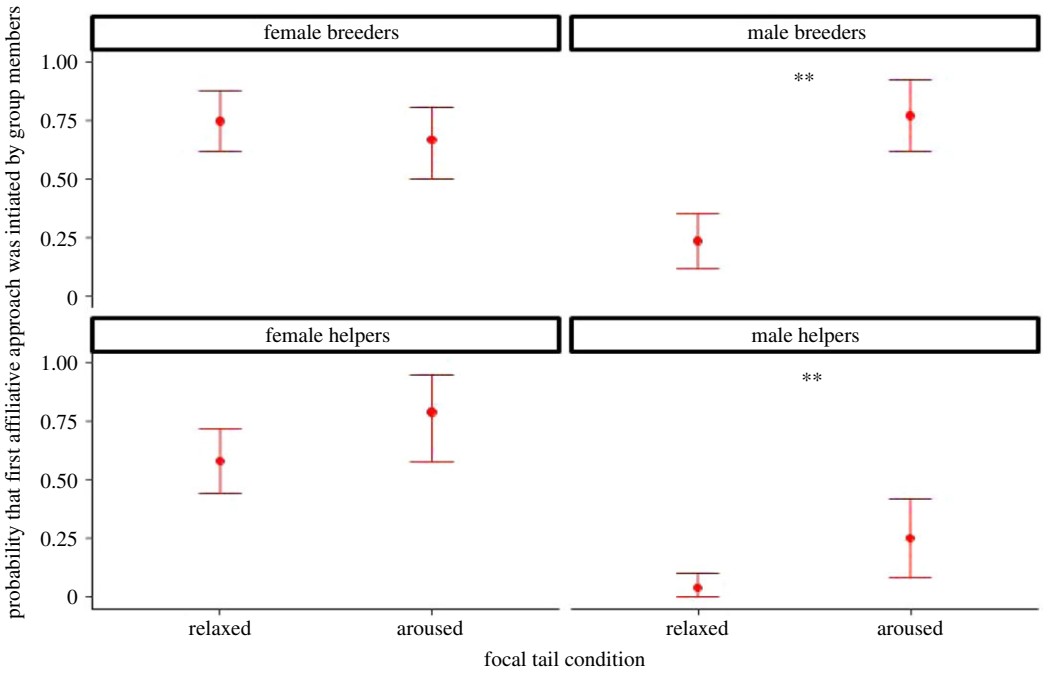

**Figure 4.** Probability that first affiliative contact was initiated by group members, split by focal subject sex-status and according to its tail condition in first 3 min. *Post hoc* pairwise tests; **: $p < 0.01$ (estimate marginal means $\pm$ 1 s.e.m., $N = 16$).

**Table 4.** ANOVA table on group member's affiliative interactions towards focal subjects. Italics, $p < 0.005$.

| fixed factors | $\chi^2$ | d.f. | *p*-value |
|---|---|---|---|
| *model 4*: affiliative interaction | | | |
| group member tail condition | 2.307 | 1 | 0.129 |
| focal tail condition | 7.727 | 1 | *0.005* |
| focal subject sex | 0.246 | 1 | 0.619 |
| focal tail condition * focal subject sex | 4.309 | 1 | *0.038* |
| full model ($R^2m = 0.316$; $R^2c = 0.977$) | 59.474 | 4 | *<0.001* |

that would also include the group member's arousal, we provide establishing the first affiliative contact for all focal subjects from both relaxed and aroused group members in the electronic supplementary material, results D and figure S3.

Next, we examined *C. jacchus*' first affiliative behaviours after the reunion, and whether they were executed by the focals or the group members. We found that focal subjects indeed received more affiliation when aroused, and for both male and female focals ($\chi^2 = 59.474$, d.f. = 4, $p < 0.001$—table 4; see also electronic supplementary material, results D and table S8 for pairwise comparisons and figure S4). Although a three-way interaction model with our current dataset that would also include group member's arousal is not viable, this behaviour was not only shown by aroused group members (as expected for reassurance seeking) but also by relaxed ones (figure 5).

## 4. Discussion

We investigated marmoset monkeys' consolation behaviour and empathic abilities by studying the responses of relaxed monkeys when they came back from an outdoor enclosure and encountered an aroused focal group member (i.e. a fellow group member showing piloerection of the tail). The returning group members matched the arousal of the focals and showed piloerection themselves after having encountered an aroused conspecific, but not after having encountered a relaxed one. They, therefore, showed a matching response that was robust over all classes of animals (breeders and

**Figure 5.** Probability that first affiliative interactions were initiated by group members, split by group member's tail condition and focal subject sex, according to its tail condition in first 3 min. *Post hoc* pairwise tests: relaxed versus aroused females: $p = 0.011$, relaxed versus aroused males: $p = 0.023$; $N = 16$.

helpers of both sexes), i.e. contagion of emotional arousal, which is one of the key elements of the combination model of empathy. Moreover, for male focals, the first affiliative contacts were more likely initiated by group members than by focals when the latter were aroused rather than relaxed. Affiliative behaviours were more likely initiated by group members when the focals were aroused for both male and female focals (see electronic supplementary material, results D, figure S4 and table S8). Importantly, these consolation behaviours were performed by group members who were both themselves aroused and relaxed. In fact, establishing affiliative contact, was tendentially provided more by relaxed group members (male breeders and female helpers—figure S3 in electronic supplementary material, results D). These particular cases are of notable relevance because they can exclude that the group members who engaged in these behaviours were merely seeking for reassurance to ease their own arousal.

These results thus complement our understanding of marmoset empathic competence, which we discuss in light of the framework of the combination model [15,21]. Figure 6 summarizes our current knowledge regarding empathic competence in marmosets. Together with the current results of this study, we now have solid evidence that marmosets possess basic competencies in all three of the overlapping, yet independent components. They thus meet all preconditions for displaying one of the behaviours at the centre of the model, i.e. empathy-driven consolation. Therefore, the affiliation behaviours that aroused subjects receive from their group members ([30], and this study) may indeed represent empathy-driven consolation, i.e. a prosocial effort to reduce the partner's arousal.

The experimental manipulation reliably induced emotional arousal (piloerection of the tail) in the focal subjects, and displaying emotional arousal led to a matching response when the returning group members encountered the focal individual. The matching response occurred across all classes of animals. Thus, the group member's probability of piloerection during the first 3 min was higher after witnessing an aroused focal subject than a non-aroused one, and this pattern also held when separately looking at all focal sex-status classes (figure 3) as well as all group member's sex-status classes (see also electronic supplementary material, results C and table S4). These results complement the findings by Massen *et al.* [47] and Watson *et al.* [46] and together provide strong evidence for contagion of emotional arousal in common marmosets.

The consolation response was analysed with regard to who would establish the first affiliative contact after reunion, and with regard to the direction of affiliative interactions. We found that group members

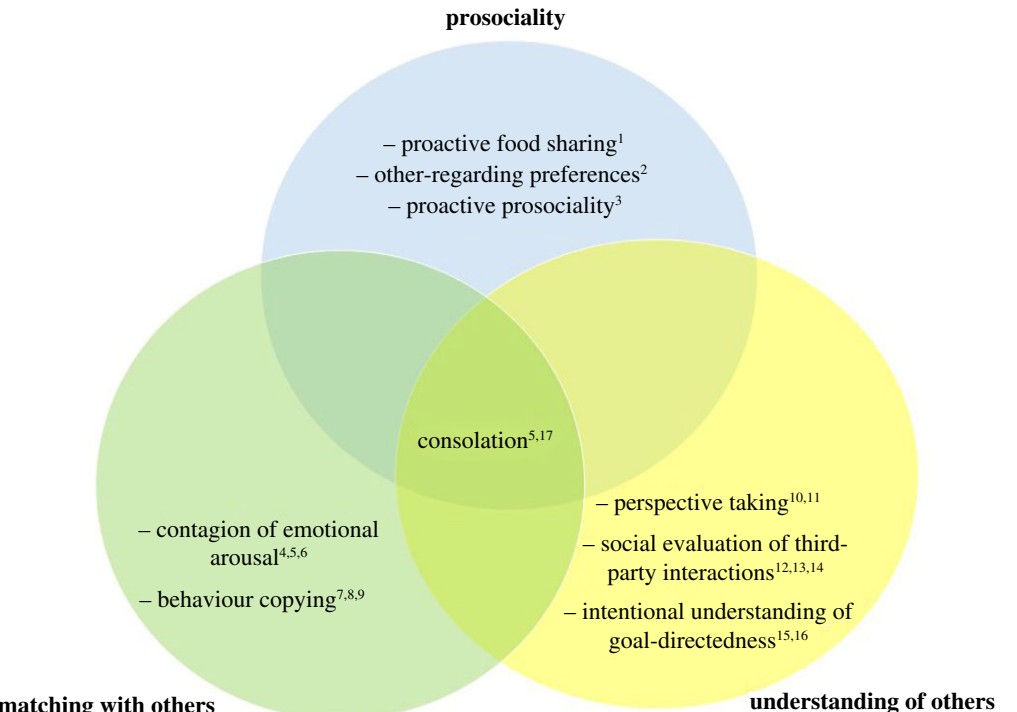

**Figure 6.** The combination model of empathy (adapted from Yamamoto [21] and Adriaense et al. [15]), and evidence from marmosets for the corresponding components. [1]Brown et al. [34]; [2]Burkart et al. [24]; [3]Burkart et al. [49]; [4]Massen et al. [47]; [5]this study (2021); [6]Watson et al. [46]; [7]Bugnyar & Huber [44]; [8,9]Voelkl & Huber [43,45]; [10,11]Burkart & Heschl [38,50]; [12,13]Kawai et al. [37,51]; [14]Brügger et al. [52]; [15]Burkart et al. [35]; [16]Kupferberg et al. [36]; [17]de Boer et al. [30].

were more likely to establish contact with the focal when focals were aroused rather than when not, and when the focal was a male. Likewise, focals were more likely to be recipients, rather than donors, of affiliative behaviours such as grooming and food sharing, which was true for both male and female focals. Notably, this pattern was observed from group members who themselves did and did not show signs of arousal, and in fact tended to be stronger in non-aroused group members when the focals were male breeders or female helpers. The responses of the group members who were not themselves aroused are particularly important because these cases exclude the alternative explanation that the affiliative behaviours were performed to ease their own, rather than focal individual's arousal.

Overall, sex differences were not highly pronounced, but where apparent they suggest that males are more proactive in consolation behaviours (see also electronic supplementary material, results D and table S7). This is consistent with the marmosets natural behaviour. Yamamoto et al. [29] have demonstrated the existence of diverging strategies within a common marmoset group, which could be broadly summarized as a stronger tendency toward cooperation between males and toward competition between females. The consequences of these strategies can be evident in various behavioural contexts, such as food sharing [32], infant carrying [53], reproductive competition [27] and social tolerance [55–56]. Male breeders usually play a particularly prominent role in infant care, especially in infant carrying and food sharing [28,59–60]. These social features make male marmosets particularly valuable to both aspiring female breeders as well as current female breeders and as cooperation partners for other males [61]. However, the sample size supporting these sex differences was limited, and a socio-ecological interpretation of these patterns would strongly benefit form additional data on consolation behaviour from wild-living common marmosets with a more diverse range of group compositions.

Our results add to the increasing body of evidence that having a particularly large brain, with the accompanying computational power [49] is not a critical precondition for the emergence of empathic competence. The presence of empathy-based consolation has arguably been established for the rather large-brained *Canis Lupus* [8], *Corvus corax* [9,10], *Elephas maximus* [11], *Pan troglodytes* [1], *Pan paniscus* [2] and *Homo sapiens*, but also the smaller-brained prairie voles [17] and now, the common marmosets (this study). What these species have in common is thus not so much having a large brain but rather that they show at least some level of either ecological or reproductive interdependence. The genus *Pan* shows levels of ecological interdependence, whereas corvids, prairie voles, Asian elephants and

common marmosets exhibit reproductive interdependence, to varying degrees. Notably, wolves and humans are examples of species showcasing both reproductive and ecological interdependence. Such interdependence, perhaps in particular in the reproductive context [62,63], may not only lower the threshold to adult–adult cooperation [64,65] but also favour the emergence of consolation. Further investigation will provide more phylogenetic scrutiny and breadth to our conclusion.

Ethics. All animals were kept in accordance with Swiss legislation. All experiments were in concordance with the ethical regulations in Switzerland from the Kantonales Veterinäramt ZH (licence #223/16, degree of severity = 0).

Data accessibility. Data used in this study as well as the code for statistical analysis can be found at: https://osf.io/dqkne/?view_only=29733e0d3bef442ca4da2315dff0e9e8.

Authors' contributions. F.E.d.O.T., A.A and J.M.B. wrote the manuscript. F.E.d.O.T. and J.M.B. designed the experimental procedure. F.E.d.O.T. collected the data. F.E.d.O.T. and E.P.W. analysed the data.

Competing interests. We declare we have no competing interests.

Funding. The research was funded by the Coordination for the Improvement of Higher Education Personnel (CAPES) and the Swiss National Science Foundation (grant no. 31003A_172979).

Acknowledgements. We thank Sandro Sehner, Rahel Brügger and Flavia Mobili for valuable input in early stages of our manuscript; Débora Louise da Cruz for help on inter-observer reliability; we also thank Gisep Bazzell for the indispensable work during the whole experimental procedure. Lastly, we deeply thank Rahel Brügger, Jessie Adriaense and Sonja Koski for thought-provoking feedback on early versions of this manuscript.

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
