## [Peer Review File · Royal Society Open Science]

Review History

Decision letter (RSOS-211255.R0)

Dear Dr de Oliveira Terceiro

The Editors assigned to your paper RSOS-211255 "Monkey see, monkey feel? Marmoset reactions toward conspecifics' arousal" have now received comments from reviewers and would like you to revise the paper in accordance with the reviewer comments and any comments from the Editors. Please note this decision does not guarantee eventual acceptance.

Please submit your revised manuscript and required files (see below) no later than 21 days from today's (ie 02-Sep-2021) date. Note: the ScholarOne system will 'lock' if submission of the revision is attempted 21 or more days after the deadline. If you do not think you will be able to meet this deadline please contact the editorial office immediately.

on behalf of Dr Oliver Schülke (Associate Editor) and Kevin Padian (Subject Editor)
openscience@royalsociety.org

Associate Editor Comments to Author (Dr Oliver Schülke):

Associate Editor

Comments to the Author:

The reviewer comment on latency between end of experimental condition for the focal animal and the onset of the observations reported here has not been used to change the text accordingly.

It is unclear why fig 2 presents data in 1-min intervals if the text says that analyses were run on 30-sec intervals.

The reviewer comment concerning the y-axis title was not used to its best benefit. The reader is left to guess what the units are on an axis that is labeled "Focal tail arousal (average)" and the figure legend does not make that clear either.

The variable "sex-status" is not explained and the reader is left to make sense of it themselves.

What justifies using aroused control subjects? It seems that this procedure defies the purpose of the experiment. If subjects in the control condition were aroused perhaps the other group members were too due to a common cause. This should be discussed - especially because the experimental approach is used to motivate the study in the first place. On line 279 for one analysis the reader is referred to the suppl for results based on experimental condition instead of subject arousal state. These results are said to be overall similar when in fact the interaction term of focal subject sex-status * exp condition is significant because males do not show the expected reaction (suppl tab 6).

268ff: By aggregating the very detailed data into a binary response over 3 minutes the temporal causality claims are jeopardized. It has to remain unclear whether “The group members display higher rates of piloerection after witnessing a focal subject who was aroused compared to a relaxed one”, because group members may have been aroused before the focal animal was. It cannot be concluded whether “returning group members reacted to aroused focal individuals by matching the arousal” or the other way around. If you chose to leave analyses as they are, the text has to be changed to remove any notion of temporal causality.

It is difficult to link the suppl material to analyses presented in the main text. If post-hoc tests are being presented, the table heading should name the model these results come from (model 1 through 4).

I agree with reviewer 1 that the representation of results in the text does not always reflect the statistical results presented in tables and figures very well. On line 276 the reader is referred to suppl table 5 for results on how the reaction to focal subject arousal is modified by group member sex and status. This table presents post-hoc tests that not seem fit to answer this question because here differences in the response between different sex/status groups are investigated and not how these groups would differ in the effects of focal arousal on the response. The model presented in the main text does not generate evidence for a non-significant interaction term of focal arousal * group member sex-status. Therefore, the interpretation is not supported that “The group members display higher rates of piloerection after witnessing a focal subject who was aroused compared to a relaxed one [...], and this response hold true [...] regardless of the group member’s sex and status [...] (See Table 5 in supplemental material).”

The significant interaction term and associated post-hoc analyses (suppl tab 7) shows that the effect of focal arousal on the probability that group members establish contact differs between female and male focals and that female focal arousal state did not systematically affect contacts. Therefore, the interpretation in the text is unsupported “We found that approaches were more likely initiated by group members when the focal individuals were aroused (piloerection) than when they were relaxed.” This is true for one sex only which should be clearly stated.

Then on line 303ff results seem mixed up again. The significant interaction term is broken down into posthoc tests in suppl tab 8 which supports the main claim that “focal subjects received more affiliation when aroused” in both focal subject sexes. Then the text goes on to discuss the role of group member arousal state on this effect without presenting statistical results to back up the claims made, i.e. a significant interaction effect of focal arousal * group member arousal. The plot in fig 5 does not help with bringing home the message that focal subject arousal affects group member initiation of affiliation with the focal – it leaves the (un-tested) impression that this goes for aroused group members only which is important when distinguishing between theoretical explanations for the behavior.

Why is N=16 post-hoc tests in all suppl figure legends, no matter whether the plots show 16 estimates or 8?

The issues with interpreting or inclusion of interaction terms in the statistical models culminates in the first paragraph of the discussion where broad claims are made that are not supported by the data as they are presented. Much more caution is needed and statements like “establishing friendly contact were provided more by relaxed group members” or later in the discussion “The matching response occurred across all classes of animals” are to be avoided if not backed up by data. Like the results section, the discussion needs a thorough revision to deal with these issues.

Editor comments:

Thanks for submitting to RSOS. As you see there are extensive editorial recommendations that we hope you can address and resubmit. Best wishes.

===PREPARING YOUR MANUSCRIPT===

===PREPARING YOUR REVISION IN SCHOLARONE===

- 1) One version identifying all the changes that have been made (for instance, in coloured highlight, in bold text, or tracked changes);
 - 2) A 'clean' version of the new manuscript that incorporates the changes made, but does not highlight them.
 - An individual file of each figure (EPS or print-quality PDF preferred [either format should be produced directly from original creation package], or original software format).
 - An editable file of each table (.doc, .docx, .xls, .xlsx, or .csv).
 - An editable file of all figure and table captions.
- Note: you may upload the figure, table, and caption files in a single Zip folder.
- Any electronic supplementary material (ESM).
 - If you are requesting a discretionary waiver for the article processing charge, the waiver form must be included at this step.
 - If you are providing image files for potential cover images, please upload these at this step, and inform the editorial office you have done so. You must hold the copyright to any image provided.
 - A copy of your point-by-point response to referees and Editors. This will expedite the preparation of your proof.

- Ensure that your data access statement meets the requirements at <https://royalsociety.org/journals/authors/author-guidelines/#data>. You should ensure that you cite the dataset in your reference list. If you have deposited data etc in the Dryad repository, please include both the 'For publication' link and 'For review' link at this stage.
- If you are requesting an article processing charge waiver, you must select the relevant waiver option (if requesting a discretionary waiver, the form should have been uploaded at Step 3 'File upload' above).
- If you have uploaded ESM files, please ensure you follow the guidance at <https://royalsociety.org/journals/authors/author-guidelines/#supplementary-material> to include a suitable title and informative caption. An example of appropriate titling and captioning may be found at https://figshare.com/articles/Table_S2_from_Is_there_a_trade-off_between_peak_performance_and_performance_breadth_across_temperatures_for_aerobic_sc_ope_in_teleost_fishes_/3843624.

Author's Response to Decision Letter for (RSOS-211255.R0)

See Appendix A.

Decision letter (RSOS-211255.R1)

Dear Dr de Oliveira Terceiro,

It is a pleasure to accept your manuscript entitled "Monkey see, monkey feel? Marmoset reactions toward conspecifics' arousal" in its current form for publication in Royal Society Open Science.

The comments of the reviewer(s) who reviewed your manuscript are included at the foot of this letter.

Please ensure that you send to the editorial office (and include in your proofed manuscript) the URL for your OSF deposition that will be accessible to readers/the public on publication - at present the version of the URL supplied appears to be the for-review (ie private) version.

on behalf of Dr Oliver Schülke (Associate Editor) and Kevin Padian (Subject Editor)
openscience@royalsociety.org

Associate Editor Comments to Author (Dr Oliver Schülke):

Associate Editor

Comments to the Author:

The authors did a great job in revising the manuscript to address all of the previous ProCB reviewer questions and my own concerns. They provide more detail on the methods, more guidance for the reader in the presentation of results concerning complicated interaction effects and a more balanced conclusion in the discussion section.

I am happy to suggest acceptance without further revision.

Appendix A

Dear Dr. Schülke

Thank you very much for your quick handling of this manuscript.

Overall, we agree with most comments made and we are sure that by addressing them our study improves significantly. We fully agree with the need of improving the link between main text and supplementary material and have now worked extensively to address this issue.

We have carefully edited some key statements mentioned in your comments in order to better explain our findings and also added further information required about our experimental procedure.

Below, we detail how we have dealt with each of your comments.

Yours sincerely

Francisco de Oliveira-Terceiro, Erik P. Willems, Arrilton Araújo, and Judith Burkart

Associate Editor Comments to Author (Dr Oliver Schülke):

Associate Editor

Comments to the Author:

The reviewer comment on latency between end of experimental condition for the focal animal and the onset of the observations reported here has not been used to change the text accordingly.

- Based on our response previously to the reviewers, I have now added a detailed account also in the text. They are highlighted in grey from line 163 to 166 and 169 to 172.

It is unclear why fig 2 presents data in 1-min intervals if the text says that analyses were run on 30-sec intervals.

- The analysis, as shown in supp table 3, were also by minute, as now indicated in the methods section (line 226). The data was collected in scans every 30 seconds. However, running the analysis using 30-second bins created 2 hurdles. First, the total number of interacting factors required an extensive amount of computational power for the analysis. Second, and most importantly, by using 30-second bin we had enough cases of no piloerection and/or no recordings (cases of no recording explained from line 198 to line 200) to make the whole model collapse. Hence, one-minute bins, although less granular, were used for both statistical analysis and visual representation.

The reviewer comment concerning the y-axis title was not used to its best benefit. The reader is left to guess what the units are on an axis that is labeled “Focal tail arousal (average)” and the figure legend does not make that clear either.

- I added a more detailed account in both the label in the y-axis and the subtitles for figures 2 and 3. Subtitle changes are highlighted in grey.

The variable “sex-status” is not explained and the reader is left to make sense of it themselves.

- This explanation is now included in lines 233-235 and highlighted in grey.

What justifies using aroused control subjects? It seems that this procedure defies the purpose of the experiment. If subjects in the control condition were aroused perhaps the other group members were too due to a common cause. This should be discussed - especially because the experimental approach is used to motivate the study in the first place. On line 279 for one, analysis the reader is referred to the suppl for results based on experimental condition instead of subject arousal state. These results are said to be overall similar when in fact the interaction term of focal subject sex-status * exp condition is significant because males do not show the expected reaction (suppl tab 6).

-We would like to restate here that the subjects were *not* aroused in the feeding (control) condition. Neither before nor during the releasing moment. We now say this more explicitly from line 169 to 172.

- Regarding experimental condition in the supplemental material, we have adjusted the main text (line 294 to 295) to be more precise. We now detail how the experimental condition results follow the same direction as the focal arousal result and how this is more evident among female subjects.

268ff: By aggregating the very detailed data into a binary response over 3 minutes the temporal causality claims are jeopardized. It has to remain unclear whether “The group members display higher rates of piloerection after witnessing a focal subject who was aroused compared to a relaxed one”, because group members may have been aroused before the focal animal was. It cannot be concluded whether “returning group members reacted to aroused focal individuals by matching the arousal” or the other way around. If you chose to leave analyses as they are, the text has to be changed to remove any notion of temporal causality.

- As suggested also in the comment above, we have now included a more detailed account about the latency period in the main text (lines 163 to 166 and 169 to 172), which clarifies that in the beginning of teasing condition sessions, the focals were aroused but not the returning group members. In this paragraph we now give a more detailed account of how we control for both focal and group member’s piloerection status at the beginning of the experiment, which resolves this issue.

It is difficult to link the suppl material to analyses presented in the main text. If post-hoc tests are being presented, the table heading should name the model these results come from (model 1 through 4).

- We have now improved the readability and “linkability” of the supplemental material by structuring it more clearly. Regarding the link between supplemental material and main text, we adhered to the editor’s suggestion in 2 ways. 1) we added in the main text the name of its respective section in the supplemental material whenever was the case. 2) we added in the supplemental material the respective model for each figure and table whenever was the case. We believe that this more explicit approach ought to make the connection between main text and supplemental material easier.

I agree with reviewer 1 that the representation of results in the text does not always reflect the statistical results presented in tables and figures very well. On line 276 the reader is referred to suppl table 5 for results on how the reaction to focal subject arousal is modified by group member sex and status. This table presents post-hoc tests that not seem fit to answer this question because here differences in the response between different sex/status groups are investigated and not how these groups would differ in the effects of focal arousal on the response. The model presented in

the main text does not generate evidence for a non-significant interaction term of focal arousal * group member sex-status. Therefore, the interpretation is not supported that “The group members display higher rates of piloerection after witnessing a focal subject who was aroused compared to a relaxed one [...], and this response hold true [...] regardless of the group member’s sex and status [...] (See Table 5 in supplemental material).”

- Please note that the reference in the text is to supp table 4, rather than 5. That aside, at some point over the reviews someone from the journal asked to remove this info from supp table 4. Luckily, I still had that info and added back in the supp. I also added the info in the main text and highlighted in grey. Line 291.

The significant interaction term and associated post-hoc analyses (suppl tab 7) shows that the effect of focal arousal on the probability that group members establish contact differs between female and male focals and that female focal arousal state did not systematically affect contacts. Therefore, the interpretation in the text is unsupported “We found that approaches were more likely initiated by group members when the focal individuals were aroused (piloerection) than when they were relaxed.” This is true for one sex only which should be clearly stated.

- We agree with the editor. The sentence is now adjusted and highlighted in grey.

Then on line 303ff results seem mixed up again. The significant interaction term is broken down into posthoc tests in suppl tab 8 which supports the main claim that “focal subjects received more affiliation when aroused” in both focal subject sexes. Then the text goes on to discuss the role of group member arousal state on this effect without presenting statistical results to back up the claims made, i.e. a significant interaction effect of focal arousal * group member arousal. The plot in fig 5 does not help with bringing home the message that focal subject arousal affects group member initiation of affiliation with the focal – it leaves the (un-tested) impression that this goes for aroused group members only which is important when distinguishing between theoretical explanations for the behavior.

- We fully agree with the editor on the relevance of this topic in our manuscript. Unfortunately, we cannot run a 3-way interaction (focal sex*focal arousal * group member arousal) from our dataset. We try to address this in two ways. First, from line 309 to 311 we explicitly state that we unfortunately cannot perform a three-way analysis. Second, we try to clarify how despite this, we still find a greater tendency from affiliative interactions with aroused (particularly male) focals (line 306 to 308; line 340 to 344, lines 375 and 376).

Why is N=16 post-hoc tests in all suppl figure legends, no matter whether the plots show 16 estimates or 8?

- I think this point was a small misunderstanding. N=16 referred to the overall sample size for all tests. Trying to avoid this confusion we moved the term “post hoc pair-wise tests” to another place withing each subtitle.

The issues with interpreting or inclusion of interaction terms in the statistical models culminates in the first paragraph of the discussion where broad claims are made that are not supported by the data as they are presented. Much more caution is needed and statements like “establishing friendly contact were provided more by relaxed group members” or later in the discussion “The matching response occurred across all classes of animals” are to be avoided if not backed up by data. Like the results section, the discussion needs a thorough revision to deal with these issues.

- About these 2 quotes:

“Establishing friendly contact were provided more by relaxed group members”

We think that with the changes we now implement (line 339 to 342) we are able to “tone it down” our statement by being (1) more precise about the extent of our affirmation, and (2) more specific about the subjects involved in the referred process.

“The matching response occurred across all classes of animals”

We think that we now provide enough support for this statement by adding the required data in supplemental figure 4 and by adjusting lines 288 and 289.